# Impact of Formal Care Use on Informal Care from Children after the Launch of Long-Term Care Insurance in Shanghai, China

**DOI:** 10.3390/ijerph17082938

**Published:** 2020-04-24

**Authors:** Huimin Zhang, Xiaoyi Zhang, Youhua Zhao, Jianfeng Huang, Wenwei Liu

**Affiliations:** 1School of International and Public Affairs, Shanghai Jiao Tong University, Shanghai 200030, China; zhanghuimin@sjtu.edu.cn (H.Z.); xyzhang@sjtu.edu.cn (X.Z.); jf_huang@sjtu.edu.cn (J.H.); 2School of Political Science and Public Administration, East China University of Political Science and Law, Shanghai 201620, China; 18773222567@sina.cn; 3College of Philosophy, Law and Political Science, Shanghai Normal University, Shanghai 200234, China

**Keywords:** informal care, formal care, long-term care insurance, Shanghai, China

## Abstract

The impact of formal care (co-paid by long term care (LTC) insurance) on informal care is critical to the improvement and promotion of public policy. We conducted an interview-based survey to examine how the use of formal care impacts the use of informal care in Shanghai, which was one of China’s first long-term insurance pilots in 2016. In addition to total informal care time, the following four types of informal care were considered: (1) household activities of daily living (HDL) tasks, (2) activities of daily living (ADL) tasks, (3) instrumental activities of daily living (IADL) tasks, and (4) supervision tasks. Of the 407 families, an average of 12.36 h (SD = 6.70) of informal care was crowded out each week. Among them, ADL tasks, HDL tasks, and supervision tasks were reduced an average of 4.60 (SD = 3.59), 5.50 (SD = 3.38), and 2.10 h (SD = 3.06) per week, respectively. Each additional hour of formal care reduced 0.473 h of informal care. Care recipients’ gender and health status were also determined to be associated with crowding out hours of informal care. These findings can be utilized as empirical evidence for decision-makers to consider the scope of funding for formal care, and this study provides comparable results to developing countries and regions.

## 1. Introduction

China has the largest population in the world, and 241 million people were aged over 60 years in China at the end of 2018 [1]. In addition, the population over the age of 60 will exceed 400 million in 2050, accounting for more than 30% of the total estimated population [2]. With the rapid growth of the elderly population, it is anticipated that the need for long-term care may increase substantially [3].

China started piloting long-term care (LTC) insurance on 27 June 2016 [4]. Shanghai was one of the first pilot cities in 2016, and LTC insurance was implemented in the entire city of Shanghai on 1 January 2018 [5]. As of 31 December 2018, Shanghai had a population of 5,032,800 people aged 60 and over, comprising 34.4% of the total population [6]. Indeed, Shanghai is the most aging city in mainland China. To some extent, it represents the developing trends of Chinese cities’ aging and LTC demands. LTC insurance in Shanghai is funded by the medical insurance system, in which individuals who use formal care from LTC insurance only need to bear 10% of the cost, and the medical insurance pays the remaining 90%. In addition, each elderly person can be funded for up to 7 h of formal care per week with LTC insurance. 

The demand for LTC by the frail elderly can be satisfied by formal care and/or informal care. Informal care generally refers to care activities provided by children, spouses, relatives, and friends. Formal care includes care provided by LTC insurance, community health and social services, and other paid help [7]. The most common form of LTC is informal care provided by the elderly’s adult children [8], and children account for approximately half of all informal care for older people [9]. It has been demonstrated that long-term caregivers face substantial psychological and/or health problems [10], as well as reduced employment opportunities [11] and income [12]. Governments have actively developed formal care policy systems to support LTC. A United States study demonstrated that the increased use of formal care (funded by a Medicaid home care grant) decreased the amount of informal care [13]. An investigation from Canada found that publicly-financed formal home care can result in a decline of informal care, and improve caregivers’ health status [14]. Other researchers, however, have shown that public home care program generosity had only modest, or even no, impact on the provision of informal care [15,16]. Overall, whether publicly supported LTC can reduce the burden on caregivers has an important impact on the implementation and improvement of policies in the future.

Currently, few studies exist concerning the LTC insurance policy in China have been published. Especially, there is no extant literature on the impact on various activities within informal care. However, China’s experience has important implications for the improvement and promotion of LTC public policy, and LTC public policies must consider these issues in order to produce more progressive LTC policy outcomes. Therefore, this research aims to elucidate the associations between the use of formal care and the change of total informal care time, as well as the time variation of the four classification activities, and to explore the associated factors. This study may provide comparable results for developing countries and regions. The rest of the paper is structured as follows. The method section outlines our research design, variables, data collection, and statistical methods. The results of analyses are presented in the results section, including description, univariate analysis, and multivariate analysis. The discussion of our findings, including the limitations of this research and the conclusions, are also provided, respectively.

## 2. Method

### 2.1. Design and Sample Selection Criteria

Fifty-five subdistricts from 105 subdistricts in Shanghai were randomly selected. Families in each subdistrict who met the research requirements were invited to participate in the survey. All invited families included an elderly person who had used formal care provided by long-term care (LTC) insurance for 1 to 3 months (from 15 May 2019 to 15 August 2019) and one child who is primarily responsible for the elderly person’s daily informal care. These dependent elderly people were all at least 60 years old, had no cognitive impairment, and had never used other formal care.

In total, 550 eligible families participated in the survey voluntarily. Among the 550 samples obtained, 26 cases (4.73%) were excluded due to incompletion of data, 27 cases (4.9%) were excluded for over-reported carehours (over 40 h of extrusion time per week), 13 cases (4.9%) were excluded because the informal caregiver was younger than 18 years old, 39 cases (7.1%) were excluded for too many missing values, and 38 cases (6.9%) were excluded for extreme values. After sample selection, a total of 407 valid samples were obtained.

### 2.2. Dependent Variables: Time Changes in Informal Care

Based on previous literature [17,18,19], we distinguished the following four activities as proxies for informal care: (1) household activities of daily living (HDL) tasks, (2) activities of daily living (ADL) tasks, (3) instrumental activities of daily living (IADL) tasks, and (4) supervision tasks. Specifically, HDL tasks comprised activities such as cleaning and laundry, while ADL tasks indicated activities such as personal care. IADL tasks comprised activities such as handling finances or administration, whereas supervision tasks indicated activities such as looking after the care recipient, i.e., situations in which the elderly cannot be left alone.

We asked the caregivers the following questions, “Has your time spent on HDL tasks changed since your parents received formal care co-paid by the LTC insurance? If so, how much has changed each week?”. In the same way, we also asked the caregivers if the time spent on other tasks, such as ADL, IADL, and supervision, had changed. If there was a change, we then inquired how much change occurred each week. Finally, we summed all of the change times of the four activities as the total time impact of the caregiver.

### 2.3. Independent Variables

Independent variables included the following: (1) hours of formal care (this article only analyzes the utilization of formal care co-paid by LTC insurance). LTC insurance provides formal care with 48 care items that cover HDL, ADL and IADL tasks. According to our survey, when the elderly use the formal care provided by LTC, they generally use multiple services, such as HDL care and IADL, at the same time. (2) The characteristics of the caregiver and the care recipient as covariates (i.e., gender, age, marital status, number of years of education, monthly income, and health status (self-assessment)). Health status (self-assessment) was further divided into five levels: (1) very good, (2) good, (3) fair, (4) poor, and (5) very poor. In addition, we also measured care recipients’ co-residents (whether or not the parent lives with children) and the number of living sons and/or daughters. The caregiver reaction assessment (CRA sub1–5) was also sub-measured as follows: CRA sub1—caregivers’ esteem: resent having to care (reverse), caring makes me feel good, and enjoy caring; CRA sub2—impact on health: tired all the time, health had gotten worse, healthy enough to be cared for; CRA sub3—lack of family support: others abandon caring, family left me alone, difficult to obtain help; CRA sub4—impact on finances: finances were inadequate, financial strain on family, difficult to pay; CRA sub5—impact on schedule: interruptions, activities centered on care, stop work to care.

### 2.4. Data Collection 

Following informed consent processes, nine trained investigators visited 550 families. As mentioned previously, each interviewed household included an elderly person and a child who mainly cares for the elderly person. 

All invited caregivers and care recipients consented to participate, and each respondent provided written or oral consent. Trained data collectors completed the interview-based survey, which took 30–40 min with each family to complete. In order to avoid the caregiver exaggerating his or her care burden, we first interviewed the caregiver alone in a room, without the presence of the care recipients, and then interviewed the elderly. The survey was performed between 1 September to 27 December 2019.

### 2.5. Statistical Analysis 

We used numbers and percentages for descriptive statistics for the categorical variables and used means and medians for descriptive statistics for metric variables. The significance of differences was tested by Student’s *t*-test, one-way analysis of variance (ANOVA), and Spearman’s correlation test, accordingly. Variables that had a significant association with the dependent variable in the univariate study were introduced to the multiple regression analysis. The multivariate regression model was performed to identify the influencing factors of the dependent variable. SPSS 22.0 (IBM, Armonk, NY, USA) was utilized to perform statistical analyses. *p*-Values < 0.05 were considered statistically significant.

## 3. Results

### 3.1. Characteristics of the Study Sample 

In total, 407 care recipients and 407 caregivers participated in the survey (Table 1). The mean age of care recipients was 81.59. Females constituted 63.1% of care recipients. Briefly, 33.9% of them were widowed, 61.9% were married, and only 1% were unmarried or divorced. Of the surveyed care recipients, 32.7% reported 7–9 years of formal education, followed by 30% who reported six or fewer years of formal education, 8.6% reported having a high school education, and 5.9% reported having at least a college education. The participants’ monthly incomes of <4000 yuan, 4000–6000 yuan, and >6000 yuan accounted for 48.9%, 40.8%, and 8.4%, respectively. The majority of care recipients lived independently (73%), and just 27% lived with children. The care recipients, on average, had one son and one daughter. Mean health status (self-assessment) was 3.89, with an average of 3.88 h of formal care per week.

Caregivers’ average age was 54.5, with 40.0% of men and 47.9% of women. Of them, 81.6% were married, 3.4% were widowed, and 4.4% were unmarried or divorced. Most caregivers had 10–12 years of education (33.4%), followed by 28% who reported 13 or more years of formal education, 23.1% who reported having at least some college education, and only 4.2% of caregivers reported having 6 years or less of education. Of the caregivers, monthly incomes of <4000 yuan, 4000–6000 yuan, and >6000 yuan accounted for 33.9%, 37.3%, and 14.7%, respectively. Caregivers’ self-assessed health average was 2.77.

### 3.2. Univariate Analysis

An average of 23.40 h per week (SD = 11.2) of informal care is provided, and the crowd-out time is presented in Table 2, according to the task presented. Briefly, 90.4% of caregivers reported a decrease of total informal care time, and only 2.7% of caregivers reported no changes (missing = 6.9%). As shown in Table 2, after using formal care, caregivers could save an average of 12.36 h per week (SD = 6.7). HDL task time, ADL task time, IADL task time, and supervision time were, respectively, reduced by 4.6 (SD = 3.59), 5.5 (SD = 3.38), 0.75 (SD = 1.4), and 2.1 h (SD = 3.06) per week, on average.

Among the significant explanatory variables of total time, 11.32 h (SD = 6.87) less informal care per week was found when the caregiver was male, while this was 12.99 h (SD = 6.55) per week for females. In addition, care recipients’ health status and hours of formal care time’s Spearman rank correlation coefficients with total time were 0.188 (*p* < 0.001) and 0.498 (*p* < 0.001), respectively. The correlation coefficients of caregivers’ health status and CRA with total time were 0.017 (*p* = 0.017), −0.151 (CRA sub1, *p* = 0.003), −0.201 (CRA sub2, *p* < 0.001), −0.109 (CRA sub3, *p* = 0.034), 0.180 (CRA sub4, *p* < 0.001), and 0.230 (CRA sub5, *p* < 0.001), respectively.

Among the explanatory variables of the four care activities, hours of formal care constituted an important explanatory variable and exerted a significant impact on HDL task time (*p* < 0.001), ADL task time (*p* < 0.001), and supervision (*p* = 0.007). Moreover, care recipients’ gender (*p* = 0.013) and marital status (*p* < 0.001) had a significant effect on HDL task time and IADL task time, respectively. Both care recipients’ age (*p* = 0.026) and health status (*p* < 0.001) significantly affected ADL task time. Caregivers’ gender, marital status, and years of education imparted a significant impact on IADL task time (*p* = 0.002), HDL task time (*p* = 0.003), and ADL task time (*p* = 0.041), respectively. Caregivers’ income level was also a significant variable for ADL task time (*p* = 0.045), IADL task time (*p* = 0.025), and supervision (*p* = 0.002). Furthermore, the caregiver reaction assessment (CRA sub1–5) exhibited had varied significant impacts on the four care activities.

### 3.3. Associations of Four Care Activities with Relevant Variables Based on Multiple Linear Regression 

After we investigated the demographic and related variables of care recipients and caregivers, we found that the hours of formal care and some characteristics exerted an impact on the time of caregivers (Table 3). Specifically, multivariate linear regression demonstrated that the longer formal care was used, the less was the total informal care time (standardized β = 0.473, 95% CI: 0.371–0.563), and the less was the HDL task time (standardized β = 0.21, 95% CI: 0.112–0.327), ADL task time (standardized β = 0.509, 95% CI: 0.416–0.611), and supervision time (standardized β = 0.157, 95% CI: 0.043–0.267).

In addition, in the case of the poor health of the caregiver, the use of formal care decreased both in the total time of informal care (standardized β = 0.108) and ADL task time (standardized β = 0.16). Compared with high-income caregivers, low-income caregivers reduced less IDL tasks and supervision tasks time but reduced more time in ADL tasks. Gender also influenced the utilization of formal care. When the care recipients were female, the total informal care time (standardized β = 0.119) was more reduced, whereas when the caregivers were male, IADL task time (standardized β = −0.182) was reduced to an even greater extent. Regarding the impact of CRA on informal care, the more caregivers became tired of caring activities (sub 2), the more crowded out was the supervision time (standardized β = −0.185). Moreover, when informal caring activities had more family support (sub 3), the addition of formal care allowed caregivers to save more time in IADL tasks (standardized β = −0.223). In addition, the greater the financial pressure that care activities placed on families (sub 4), the more total informal care time (standardized β = 0.093) and HDL task time (standardized β = 0.127) was reduced. Furthermore, when caring activities had a greater impact on caregivers to work (sub 5), the more crowded out was ADL task time (standardized β = 0.097). Other associations were not statistically significant.

## 4. Discussion

### 4.1. Primary Findings

This study examined the association of using formal care co-paid by LTC insurance with the change of informal care use provided by children in Shanghai, China. Our research confirms that the utilization of formal care was significantly associated with the change in informal care when controlling for caregivers’ and care recipients’ characteristics. On average, elderly people in our sample received 3.88 h per week of formal care co-paid by the LTC insurance. After using formal care, an average of 12.36 h (SD = 6.70) of informal care was crowded out each week. Moreover, ADL tasks, HDL tasks, IADL tasks, and supervision time were crowded out an average of 4.60 (SD = 3.59), 5.50 (SD = 3.38), 0.75 (SD = 1.40), and 2.1 h (SD = 3.06) per week, respectively. 

The use of formal care reduced the use of informal care provided by children of the recipients. This result was in accordance with the results of previous international studies [14,15,16,20,21]. Moreover, the extent of the impact of using formal care on informal care was likely to differ according to the caregivers’ and care recipients’ characteristics, as well as care type [22]. Care recipients’ gender and health status were determined to be associated with the reduction of total informal care time. Previous research also demonstrated that women required more care and were more apt to use formal care than men [7]. Overall, increasing formal care can provide more healthcare support to people in poor health and may improve health status [7]. 

Hours of formal care and CRA sub4 (impact on finances) were shown to be significantly correlated with hours of HDL activities in informal care. Indeed, a social survey revealed that formal care can constitute supplemental work of child caregivers for personal care tasks and, to a lesser extent, housework. [7] It may also be a substitute for informal care by children because formal care provided in Shanghai covers a part of HDL tasks. In addition, when HDL tasks place greater financial pressure on children, formal support may replace more care activities by children. However, further investigations are needed to verify this assertion. In terms of impact on ADL activities of informal care, hours of formal care, care recipients’ health status, caregivers’ income, and CRA sub5 (impact on schedule) were identified as factors associated with the change of informal care hours. Formal organizations were also more suited to perform technical and routine tasks, such as nursing care [23]. An investigation from Korea also reached the same conclusion, i.e., care recipients used more professional care and reduced their use of informal care when in poor health [23]. Furthermore, when caregivers had lower incomes, more informal care time may be reduced, which may be attributable to that low-income families had more difficulty accessing professional care and health services in their daily lives [24]. Therefore, ADL activities provided by caregivers who had low-incomes were more easily supplanted by those provided by professionals. Caregiver depression was also negatively related to caregiver schedule [25]. In addition, caregivers provided informal care when there was less impact on work [21]. Indeed, it is possible that the traditional Chinese concept of family obligations may be an influencing factor.

Furthermore, the use of formal care only crowded out supervision, and no significant association with hours in IADL activities was observed. This may be because there were fewer IADL activity items covered by LTC insurance, and thus the crowding-out effect was not obvious. Higher levels of income were also correlated with an increasing extent of the crowding-out effect in IADL and supervision activities. This could probably be ascribed to that the use of informal care was substituted by purchasing other assistive devices as supplementary care. CRA sub3 (lack of family support) and caregiver gender were thought to be related to more reduction in IADL task hours. A caregiver program from Shanghai also showed that a more supportive caregiver can reduce his or her care time and pressure [26], decrease participation in care activities, and increase the use of formal care [10]. An investigation also reported that women lagged behind men in IADL tasks, such as calling and shopping, and performed less IADL care [27]. CRA sub2 (effects on health) was also considered to significantly affect changes in supervision time. A qualitative study conducted in four European countries reported that, when caregivers felt that their physical strength was greatly affected, it was difficult to carry out care activities, and supplementary assistance from others was needed [28].

### 4.2. Limitations

Our research possessed several limitations. First, our sample was limited to Shanghai, and the results can only represent Chinese cities to a certain extent. This is especially the case because copayment and service delivery methods in China’s first pilot cities may not be identical, although they may reproduce the most recent situation in Shanghai. Furthermore, the cross-sectional data were inadequate to detect weak or moderate associations, such as co-residents or different effects on the informal care time change. Finally, due to limited research resources and unavailability of sufficient participants, we did not repeat the evaluation before and after using long-term insurance, nor did we use a control group. Future research may consider employing a diary survey to collect data on the time taken by caregivers before and after formal care interventions, or set up a control group to achieve a clearer comparison.

## 5. Conclusions

The present study demonstrates that substantial informal care was substituted by formal care, which was co-paid by long-term care(LTC) insurance. ADL, HDL, and supervision tasks were mainly crowded out, but there was no obvious crowding-out effect on IADL tasks. It was also found that more social support and formal home care may reduce the level of burden of caregiving. Overall, the findings of this study clearly demonstrate that local governments in China should develop the LTC insurance system and increase the supply of formal care to more optimally distribute the pressure of informal caregiving. This study may also provide comparable results for developing countries and regions that are facing or are about to face a rapidly aging society.

## Figures and Tables

**Table 1 ijerph-17-02938-t001:** Characteristics of caregivers and care recipients.

Indepedent Variables	Care Recipients		Caregivers	
	N	%	N	%
Total	407	100	407	100
Gender	(missing = 8)		(missing = 49)	
Male	142	34.9	163	40.0
Female	257	63.1	195	47.9
Marital status	(missing = 13)		(missing = 43)	
Married	252	61.9	332	81.6
Widowed	138	33.9	14	3.4
Unmarried or Divorced	4	1.0	18	4.4
Years of education	(missing = 93)		(missing = 46)	
<=6 years	122	30.0	17	4.2
7–9 years	133	32.7	94	23.1
1–12 years	35	8.6	136	33.4
>=13 years	24	5.9	114	28.0
Income(CNY)	(missing = 117)		(missing = 57)	
<4000 yuan	122	30.0	17	4.2
4000–6000 yuan	133	32.7	94	23.1
>6000 yuan	35	8.6	136	33.4
Have children younger than 18 years old			(missing = 31)	
No			284	75.5
Yes			92	24.5
Co-residence				
No	297	73.0		
Yes	110	27.0		
	Mean	Median	Mean	Median
Age	81.59	82	54.44	56
Number of living children				
Son	1	1		
Daughter	1.04	1		
Health status (self-assessment)	3.89	4	2.77	3
Hours of formal care	3.88	3.00		
Caregiver reaction assessment				
CRA sub1			2.76	3.00
CRA sub2			2.64	3.00
CRA sub3			2.62	3.00
CRA sub4			1.16	1.00
CRA sub5			1.38	1.00

**Table 2 ijerph-17-02938-t002:** Univariate analysis.

		Crowding out of Informal Care				
Indepedent Variables		Total Time	HDL Task Time	ADL Task Time	IADL Task Time	Supervision Time
		Mean (SD)	Mean (SD)	Mean (SD)	Mean (SD)	Mean (SD)
Care recipient	Total	12.36 (6.71)	4.60 (3.59)	5.50 (3.38)	0.75 (1.40)	2.1 (3.060)
	Gender					
	Male	11.32 (6.87) *	4.00 (3.69) *	5.30 (3.30)	0.61 (1.38)	1.96 (3.15)
	Female	12.99 (6.55) *	4.98 (3.49) *	5.60 (3.39)	0.84 (1.42)	2.26 (3.02)
	Age	0.106	−0.007	0.119 *	0.012	0.106
	Marital status					
	Married	12.11 (6.93)	4.60 (3.82)	5.41 (3.50)	0.68 (1.32) ***	1.99 (3.07)
	Widowed	12.70 (6.12)	4.77 (3.21)	5.57 (3.14)	0.79 (1.34) ***	2.32 (3.02)
	Unmarried or Divorced	19.25 (10.18)	4.25 (3.40)	8.00 (2.71)	3.50 (4.04) ***	3.50 (4.04)
	Years of education					
	<=6 years	12.48 (6.92)	4.89 (3.75)	5.37 (3.53)	0.65 (1.34)	2.38 (3.17)
	7–9 years	12.31 (6.56)	4.30 (3.67)	5.69 (3.49)	0.84 (1.54)	2.06 (2.94)
	10–12 years	12.22 (7.29)	5.56 (3.65)	4.58 (3.29)	0.97 (1.35)	1.73 (2.93)
	>=13 years	13.43 (7.50)	4.53 (3.00)	4.86 (3.26)	0.71 (1.06)	3.87 (4.02)
	Income (CNY)					
	<4000 yuan	12.72 (6.38)	4.92 (3.38)	5.71 (3.39)	0.79 (1.41)	1.89 (2.91)
	4000–6000 yuan	12.09 (7.03)	4.37 (3.81)	5.11 (3.32)	0.76 (1.44)	2.51 (3.21)
	>6000 yuan	11.91 (7.22)	4.19 (3.66)	6.07 (3.34)	0.61 (1.23)	1.58 (2.89)
	Co-resident					
	No	12.26 (6.65)	4.52 (3.56)	5.44 (3.43)	0.72 (1.36)	2.16 (2.97)
	Yes	12.63 (6.89)	4.78 (3.69)	5.64 (3.31)	0.84 (1.52)	2.06 (3.30)
	Number of children					
	Sons	0.02	−0.011	0.013	0.06	0.11
	Daughters	0.027	−0.078	0.092	−0.061	0.046
	Health status (self-assessment)	0.188 ***	0.091	0.237 ***	−0.046	0.036
	Hours of formal care	0.498 ***	0.250 ***	0.562 ***	0.096	0.158 **
Caregiver	Gender					
	Male	13.02 (6.68)	4.79 (3.57)	5.86 (3.42)	1.02 (1.51) **	2.19 (3.00)
	Female	12.01 (6.77)	4.50 (3.60)	5.31 (3.39)	0.52 (1.27) **	2.20 (3.18)
	Age	−0.014	−0.093	0.096	−0.081	0.008
	Marital status					
	Married	12.56 (6.79)	4.49 (3.48) **	5.62 (3.38)	0.80 (1.44)	2.23 (3.10)
	Widowed	11.42 (4.82)	5.17 (2.52) **	5.43 (3.42)	0.45 (0.82)	2.08 (2.71)
	Unmarried or Divorced	13.56 (6.10)	7.39 (4.47) **	5.06 (3.76)	0.38 (1.26)	0.88 (2.39)
	Years of education					
	<=6 years	12.94 (6.99)	5.27 (3.94)	6.35 (4.26) *	0.40 (0.91)	1.59 (2.67)
	7–9 years	12.47 (6.25)	4.73 (3.59)	5.85 (3.26) *	0.61 (1.39)	2.10 (3.01)
	10–12 years	13.11 (6.82)	4.70 (3.52)	5.97 (3.38) *	0.81 (1.45)	2.22 (3.15)
	>=13 years	11.65 (6.62)	4.44 (3.69)	4.86 (3.30) *	0.75 (1.32)	2.02 (3.03)
	Income					
	<4000 yuan	12.01 (5.62)	4.90 (3.54)	6.02 (3.21) *	0.42 (1.11) *	1.17 (2.47) **
	4000–6000 yuan	12.43 (7.30)	4.30 (3.47)	5.61 (3.61) *	0.84 (1.52) *	2.34 (3.18) **
	>6000 yuan	12.89 (6.59)	5.18 (4.08)	4.71 (3.26) *	0.86 (0.95) *	2.67 (3.05) **
	Have children younger than 18 years-old					
	No	12.14 (6.61)	4.34 (3.48)	5.59 (3.45)	0.79 (1.44)	2.22 (3.04)
	Yes	13.13 (6.94)	5.2 (4.07)	5.2 (3.11)	0.76 (1.38)	2.17 (3.21)
	Health status (self-assessment)	0.126 *	0.069	0.194 ***	−0.071	−0.007
	Caregiver reaction assessment					
	CRA sub1	−0.151 **	−0.057	−0.163 **	−0.066	−0.137 *
	CRA sub2	−0.201 ***	0.250 ***	−0.192 ***	−0.117 *	−0.130 *
	CRA sub3	−0.109 *	0.009	−0.080	0.025	−0.119 *
	CRA sub4	0.180 ***	0.127 **	0.147 **	−0.031	0.061
	CRA sub5	0.230 ***	0.104	0.240 ***	0.045	0.079

Note: * *p* < 0.05; ** *p* < 0.01; *** *p* < 0.001.

**Table 3 ijerph-17-02938-t003:** Associations of four care activities with relevant variables based on the multiple linear regression model.

	β Coefficient	95% CI β Coefficient	SE	Standardized β Coefficient	*p*
Crowding out of total informal care time (R^2^ = 0.308; adjusted R^2^ = 0.299, *p* < 0.001)					
(Constant)	0.006	−0.086–0.097	0.046		0.905
Hours of formal care	0.467	0.371–0.563	0.050	0.473	<0.001
Care recipients’ gender	0.117	0.025–0.209	0.047	0.119	0.013
Care recipients’ health status (self-assessment)	0.112	0.010–0.214	0.052	0.108	0.032
Crowding out of HDL task time (R^2^ = 0.085; adjusted R^2^ = 0.076, *p* < 0.001)					
(Constant)	−0.019	−0.124–0.086	0.053		0.718
Hours of formal care	0.220	0.112–0.327	0.055	0.218	<0.001
CRA sub4	0.124	0.019–0.229	0.053	0.127	0.021
Crowding out of ADL task time (R^2^ = 0.386; adjusted R^2^ = 0.376, *p* = 0.044)					
(Constant)	−0.030	−0.123–0.063	0.047		0.522
Hours of formal care	0.513	0.416–0.611	0.050	0.509	<0.001
Care recipients’ health status (self-assessment)	0.170	0.070–0.270	0.051	0.160	0.001
Caregivers’ income (<4000 yuan)	0.218	0.099–0.338	0.061	0.223	<0.001
Caregivers’ income (4000–6000 yuan)	0.166	0.047–0.285	0.060	0.170	0.006
CRA sub5	0.098	0.003–0.194	0.049	0.097	0.044
Crowding out of IADL task time (R^2^ = 0.121; adjusted R^2^ = 0.109, *p* < 0.001)					
(Constant)	0.012	−0.098–0.121	0.056		0.835
CRA sub3	−0.228	−0.339–−0.116	0.057	−0.223	<0.001
Caregivers’ gender	−0.183	−0.292–−0.074	0.055	−0.182	0.001
Caregivers’ income (<4000 yuan)	−0.258	−0.394–−0.121	0.069	−0.257	<0.001
Caregivers’ income (4000–6000 yuan)	−0.142	−0.276–−0.007	0.068	−0.144	0.039
Crowding out of supervision time (R^2^ = 0.130; adjusted R^2^ = 0.118, *p* = 0.014)					
(Constant)	0.020	−0.090–0.129	0.056		0.721
Caregivers’ income (<4000 yuan)	−0.336	−0.472–−0.199	0.069	−0.320	<0.001
Caregivers’ income (4000–6000 yuan)	−0.166	−0.287–−0.045	0.061	−0.156	0.007
Hours of formal care	0.155	0.043–0.267	0.057	0.157	0.007
CRA sub2	−0.181	−0.314–−0.049	0.067	−0.185	0.007

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
