# Peer review of "Impact of Formal Care Use on Informal Care from Children after the Launch of Long-Term Care Insurance in Shanghai, China"

_ijerph, 2020, doi:10.3390/ijerph17082938_

Round 1

Reviewer 1 Report

This study surveyed (through interview) elderly people who utilized long term care (LTC) and their family member caregivers in Shanghai regarding change in informal care time after using formal care. A total of 407 pairs of elderly and caregivers were included. The interviewer collected information on sociodemographics, health status (self-assessment), and change on informal care hours. The study found a reduction of 12.36 hours in average per week after using formal care (mean 3.88 hours per week), primarily in household activities, activities of daily living, and supervision tasks. The manuscript was well written.

I have a few comments:

  1. A more robust study design would be to assess the among of informal care time before and after formal care use (as the authors discussed), or to assess two groups of elderly people who used formal care vs those who did not use formal care (controlling for health status and other factors is certainly needed). The question “has your time spent on HDL tasks change since your parents received formal care co-paid by the LTC insurance” seems leading.
  2. Was the study interview conducted after the first time use of the formal care? How long is the gap between the first formal care and the interview (time to recall)?
  3. Is the type of formal care received available? this information may help understand the specific types of “time-saving” care.
  4. What is the total number informal care hours per week?
  5. Table 2, is the number 4.6 in the total informal care time a typo?

Author Response

Response to Reviewer 1 Comments

Point 1: A more robust study design would be to assess the among of informal care time before and after formal care use (as the authors discussed), or to assess two groups of elderly people who used formal care vs those who did not use formal care (controlling for health status and other factors is certainly needed). The question “has your time spent on HDL tasks change since your parents received formal care co-paid by the LTC insurance” seems leading.

Response 1: Thanks for the suggestion.

Thank you for the valuable suggestion. The study design is a very important issue, which we have discussed many times and have attempted to implement various solutions. At first, we used the diary survey method to investigate eight families, and recorded the changes in informal care provided by the children after joining the long-term care insurance. However, the time span of this survey method is too large, and it has high requirements for investigators to apply. Next, we also considered setting up a control group, and this method has also been carried out in previous studies. However, when we screened the situation of families and the elderly who volunteered to participate in the survey, we found it difficult to set up a control group under the control of other variables. Considering the limited research resources, these two methods were not implemented ultimately. Finally, we decided to use the interview method to complete the survey. This method has also been employed in other studies, and we believe that it can serve as a better measure of changes in informal care. We have explained the limitations of our research design in the Limitations part. Please see changes in the revised manuscript. (lines 269-273)

Point 2: Was the study interview conducted after the first time use of the formal care? How long is the gap between the first formal care and the interview (time to recall)?

Response 2: Thank you for the important questions.

We did not conduct interviews after the first time of use of formal care. All invited families included an elderly person who had used formal care provided by long-term care insurance for 1 to 3 months (from May 15, 2019 to August 15, 2019). This was done to collect a sufficient number of samples. In addition, from the first use of formal care to our interviews, the amount of formal care used by the elderly did not change every week, and thus we did not re-evaluate. We have revised this in the Method part. Please see changes in the revised manuscript. (lines 77-81)

Point 3: Is the type of formal care received available? this information may help understand the specific types of “time-saving” care.

Response 3: Thank you for the helpful suggestion.

Our research only analyzes utilization of formal care co-paid by LTC insurance. LTC insurance provides formal care with 48 care items, which cover HDL, ADL, and IADL tasks. According to our survey, when the elderly use formal care provided by long-term care, they generally use multiple services, such as HDL care and IADL, at the same time. It is also for this reason that the formal care used by the elderly is difficult to measure by task. We have revised this in the Method part. Please see changes in the revised manuscript. (lines 104-108)

Point 4: What is the total number informal care hours per week?

Response 4: Thanks for the suggestion.

Thank you for the useful question. An average of 23.40 hours per week (SD=11.2) of informal care is provided, and the crowd out time is shown in Table 2 according to the task presented. We have revised this accordingly. Please see marks in the revised manuscript. (line 159)

Point 5: Table 2, is the number 4.6 in the total informal care time a typo?

Response 5: Thank you for pointing this out to us.

The number 4.6 in the total informal care time is a typo. After using formal care, caregivers could save an average of 12.36 hours per week (SD = 6.7). We have revised this in Table 2. Please see marks in the revised manuscript. (line 183)

Reviewer 2 Report

In this article, Huimin Zhang and colleagues provided their observation of the impact of formal care use on informal care from children after the launch of long-term care insurance in Shanghai, China. The major contribution of the results is to provide empirical evidence for decision-makers to consider the length of funding for formal care and provides comparable results for developing countries and regions. Generally, this manuscript is well prepared. The article is written logically and the results are clear to understand. However, there are some mistakes need further revision: In table 1, the missing case number for caregivers is wrong. The total number of income subgroups for care recipients is wrong. Several such mistakes are found. Please go through all the data to make them correct.

Author Response

Response to Reviewer 2 Comments

Point 1: In this article, Huimin Zhang and colleagues provided their observation of the impact of formal care use on informal care from children after the launch of long-term care insurance in Shanghai, China. The major contribution of the results is to provide empirical evidence for decision-makers to consider the length of funding for formal care and provides comparable results for developing countries and regions. Generally, this manuscript is well prepared. The article is written logically and the results are clear to understand.

Response 1: Thank you for your kind encouragement.

According to your suggestions, we have further revised the manuscript to improve it.

Point 2: However, there are some mistakes need further revision: In table 1, the missing case number for caregivers is wrong. The total number of income subgroups for care recipients is wrong. Several such mistakes are found. Please go through all the data to make them correct.

Response 2: Thank you for the important comment.

The number errors in Table 1 and the corresponding parts of the text have been modified and marked. We have also re-checked all of the data. Please see changes in the revised manuscript. (lines 150-156)

Reviewer 3 Report

General remarks

The paper deals with a very important issue: the association between the use of formal care and change of total informal care time. This is an area that has received little scholarly attention, especially as it relates to non-Western countries such as China. That said, a few things need tightening though to further enhance the quality of the paper.

The paper is clearly written. However, the final version should be edited.

Introduction

-The introduction is comprehensively referenced to the literature and identifies a research gap which the authors seek to fill.

-The authors say: “China had the largest population in the world…” (line 34). This should be in the present tense as China currently has the largest population in the world.

-In a sentence or two, the authors should define formal and informal care. This would be helpful to the non-specialist reader of their article.

-The authors should insert a sentence or two after line 66 that states how the paper is organized, as this will guide readers. For instance, “the rest of the paper is structured as follows. The next section outlines our research methodology. Attention then shifts to a presentation and discussion of our findings.” In other words, this should be the last sentence before you move to the next section (that is, “Method”).

Method

-Numbers should be written in words at the start of a sentence (line 69).

Results

-The authors note: “The participants’ monthly incomes of < 4000, 4000 - 6000, and > 6000 135 accounted for 48.9%, 40.8%, and 8.4%, respectively” (lines 134-135) and Lines 143-144. I assume that this is in China’s currency—Yuan. This should be made clear by the authors.

Conclusion

-The authors note: “And provides comparable results for developing countries and regions confronting with fast growing ageing population” (lines 264-265). This sentence should be rephrased in order to bring out the intended message. Additionally, the claim that this study provides comparable results is problematic as contexts are different. Not all so-called developing countries have a significant aging population.

Author Response

Response to Reviewer 3 Comments

Point 1: Introduction

The authors say: “China had the largest population in the world…” (line 34). This should be in the present tense as China currently has the largest population in the world.

Response 1: Thank you for pointing this out to us.

We have modified it to be “China has the largest population in the world…”. Please see changes in the revised manuscript. (line 34)

Point 2: In a sentence or two, the authors should define formal and informal care. This would be helpful to the non-specialist reader of their article.

Response 2: Thank you for the helpful suggestion.

We have added the definitions of formal care and informal care in the manuscript, and emphasized that this article focuses on the elderly who only use the formal care co-paid by LTC care insurance. Please see changes in the revised manuscript. (lines 48-51,104-105)

Point 3: The authors should insert a sentence or two after line 66 that states how the paper is organized, as this will guide readers. For instance, “the rest of the paper is structured as follows. The next section outlines our research methodology. Attention then shifts to a presentation and discussion of our findings.” In other words, this should be the last sentence before you move to the next section (that is, “Method”).

Response 3: Thank you for the useful suggestion.

We have added an introduction about how the paper is organized as the last sentence. Please see changes in the revised manuscript (lines 70-74)

Point 4: Method

Numbers should be written in words at the start of a sentence (line 69).

Response 4: Thank you for pointing this out to us.

We have modified it accordingly. Please see changes in the revised manuscript. (line 77)

Point 5: Results

The authors note: “The participants’ monthly incomes of < 4000, 4000 - 6000, and > 6000 135 accounted for 48.9%, 40.8%, and 8.4%, respectively” (lines 134-135) and Lines 143-144. I assume that this is in China’s currency—Yuan. This should be made clear by the authors.

Response 5: Thank you for the helpful suggestion.

We have now marked the Chinese currency, Yuan, in many places in the text where income is mentioned, and also added instructions in the Method part. Please see changes in the revised manuscript. (lines 144-156)

Point 6: Conclusion

The authors note: “And provides comparable results for developing countries and regions confronting with fast growing ageing population” (lines 264-265). This sentence should be rephrased in order to bring out the intended message. Additionally, the claim that this study provides comparable results is problematic as contexts are different. Not all so-called developing countries have a significant aging population.

Response 6: Thanks for the suggestion. Thank you for the important comment. We have revised it to be, “This study may also provide comparable results for developing countries and regions that are facing, or are about face, a rapidly aging society”. Please see changes in the revised manuscript. (lines 281-282)

Round 2

Reviewer 2 Report

The authors responded the questions accordingly.